# Unsupervised Learning from Noisy Networks with Applications to Hi-C Data

Bo Wang*, Junjie Zhu[2], Oana Ursu[3], Armin Pourshafeie[4], Serafim Batzoglou[1] and Anshul Kundaje[3,1]

[1]Department of Computer Science, Stanford University
[2]Department of Electrical Engineering, Stanford University
[3]Department of Genetics, Stanford University
[4]Department of Physics, Stanford University

## Abstract

Complex networks play an important role in a plethora of disciplines in natural sciences. Cleaning up noisy observed networks poses an important challenge in network analysis. Existing methods utilize labeled data to alleviate the noise the noise levels. However, labeled data is usually expensive to collect while unlabeled data can be gathered cheaply. In this paper, we propose an optimization framework to mine useful structures from noisy networks in an unsupervised manner. The key feature of our optimization framework is its ability to utilize local structures as well as global patterns in the network. We extend our method to incorporate multi-resolution networks in order to add further resistance in the presence of high-levels of noise. The framework is generalized to utilize partial labels in order to further enhance the performance. We empirically test the effectiveness of our method in denoising a network by demonstrating an improvement in community detection results on multi-resolution Hi-C data both with and without Capture-C-generated partial labels.

## 1   Introduction

Complex networks emerge in a plethora of disciplines including computer science, social sciences, biology and etc. They entail non-trivial topological features and patterns critical to understanding interactions within complicated systems. However, observed networks from data are typically noisy due to imperfect measurements. The adverse effects of noise pose a critical challenge in unraveling clear structures and dynamics in the networks. Therefore, network denoising can strongly influence how the networks are interpreted, and can significantly improve the outcome of down-stream analysis such as global and local community detection.

The goal of community detection is to identify meaningful structures/communities underlying the provided samples in an unsupervised manner. While the performance of community detection algorithms can worsen due to noise [1], one may use prior knowledge about the structure of the communities, such as the presence of clusters, to recover local networks [25]. In addition to the special structure that one may expect in a network, a small portion of high confidence links may be available. The combination of the special structure and the confident links can be used to denoise the network that might include both noisy or missing links. How to incorporate multiple sources of information to construct a network has been widely studied in the context of data fusion or data aggregation [3].

Biology offers a special case where overall structure of the network of interest might be known from the science but the data may be riddled with noise. One example of this is the 3D structure, or folding, of DNA. In biology, this structure is important as, among other things, the DNA topology

has been shown to have a fundamental impact on gene expression in biological processes [4]. For example, many genes are in 3D contact with genomic regions that are far away in the linear genome but close in 3D space. These genomic regions contain regulatory elements that control when the gene is active [5, 6, 34]. The rules by which regulatory elements come in contact with their target genes are still unclear [7]. While the exact mechanism for this selective interaction between the regulatory elements and the target genes is unknown, the 3D organization of the genome in domains of interaction seem to play a crucial role. Furthermore, topologically associated domains (TADs) [8], where local interactions are observed, are of potential biological interest as they have been shown to be conserved between mice and humans, which suggests an ancient root for higher order structures in the genome [8].

The interaction map of regulatory elements in a genome can be viewed as a network, where each node is a regulatory element and each link represents the interaction strength between two of the elements. In this context, we not only have prior knowledge about the types of structures in this interaction network, but we also have various types of noisy and incomplete observations of the links in this network based on recently-developed technologies.

One approach to observe these links is Hi-C, an experiment which uses high-throughput sequencing to construct a 2D contact map measuring the frequency with which pairs of genomic regions co-localize in 3D space. The results of Hi-C experiments can be summarized at multiple resolutions (lower resolutions are obtained by binning genomic regions together), ranging from 1 kb to 1 Mb[8–10]. Higher resolution maps capture more fine grained location interactions but the background noise generated by random collisions can be larger [11]. Lower resolution maps have less noise, but at the cost of losing the exact localization of genomic contacts. In addition to Hi-C, there are other experiment variants such as 4C, 5C and Capture-C technologies, which provide a new window into detection of a small number of interaction links with high confidence [12–15], by focusing sequencing resources on a selected subset of contacts. For these experiments the increased confidence comes at the cost of not measuring the full contact map. Thus, the integration of multi-resolution noisy Hi-C data with high-confidence Capture-C data is not only an interesting problem in the context of general network densoising but it is also biologically relevant.

## 1.1 Related Work

Many applications in biology utilize multiple measurements to construct a single biological network. General approaches such as [16] have relied on specific models to reveal structures from the multiple measurements. However, some biological networks do not fit these model assumptions (e.g. Gaussian). For example, while Hi-C data can be summarized at multiple resolutions, standard model assumptions are not appropriate for combining the resolutions.

Furthermore, one may acquire a small subset of highly confident measurements. In the case of Hi-C data, this can be done through Capture-C [12, 13, 15] technologies. While matrix completion is a well studied problem [2] to recover missing measurements, the setting with Capture-C is slightly different. In particular, the number of highly confident entries for the $n \times n$ adjacency matrix of rank $r$ may be less than $nr \log n$, which is suggested for matrix completion [2]. Additionally, such a method would not take advantage of the large amount of data available, albeit with higher noise, from different sources.

A common application of a denoised networks is to more reliably detect biologically-relevant communities. General community detection methods have been used to find protein complexes [17, 18], genetically related subpopulations [19], like-minded individuals in a social network [20], and many other tasks [21, 22]. Aside from the all-purpose algorithms mentioned above, there are specialized algorithms for the problem of community detection in Hi-C data. Rao et al. define TADs using the specialized Arrowhead algorithm [10]. Cabreros et al. used a mixed-membership stochastic block model to discover communities in Hi-C data [23]. Their method can detect the number of communities or can be forced to find a specified number of communities. Dixon et. al defined a directionality index that quantifies the asymmetry between the upstream and downstream interaction bias for a position [8]. A hidden Markov model was subsequently used to detect biased states based on these scores [8].

## 1.2 Our Contribution

As mentioned above, Hi-C data can be represented by many different resolutions. Although the data and noise from these resolutions are not independent, the different resolutions still contain

information that can help denoise the data. We propose a model-free optimization framework to extract information from the different resolutions to denoise the Hi-C data. While generic community detection methods are limited to using only a single input network, our optimization framework is able to pool data from different resolutions, and produces a single denoised network. This framework allows us to apply community detection methods to multi-resolution Hi-C data.

Furthermore, in special cases, a subset of the interaction network may be known with a high confidence using Capture-C [12]. To our knowledge, there is no algorithm with the capability of taking advantage of this highly confident set to improve the denoising of Hi-C data. Our framework is able to take a multi-resolution network in addition to the confident set of data to denoise the corresponding network. Applying our framework to datasets with simulated ground-truth communities derived from chromosomes 14 and 21 of GM12878 in [10], we find that our framework can indeed leverage the multiple sources of information to reveal the communities underlying the noisy and missing data.

## 2 Problem Setup

### 2.1 General Learning Framework

Throughout this paper, we will use a real and symmetric $n \times n$ matrix to represent a network on $n$ nodes. Accordingly, the $(i, j)$th entry of the matrix will be used to denote the weight or intensity of a link between node $i$ and node $j$.

Suppose we want to construct a weighted network $S \in \mathbb{R}^{n \times n}$ from a noisy observation $W \in \mathbb{R}^{n \times n}$ on the same nodes, where the noise introduces false-positive and false-negative links. If the network of interest $S$ is low rank, then this inherent structure can be used to denoise $W$. This intuition that the detected noisy matrix could lie near an underlying low-rank or sparse matrix is also key to subspace detection algorithms, such as Sparse Subspace Clustering [25] and Low-Rank Representation [26]. We use this intuition to formulate our optimization framework below:

$$\text{minimize} \quad -\mathbf{tr}\left(W^\top S\right) + \lambda \, L(S, F) + \beta ||S||_F^2 \qquad \text{(OPT1)}$$

$$\text{with respect to} \quad S \in \mathbb{R}^{n \times n}, \ F \in \mathbb{R}^{n \times C}$$

$$\text{subject to} \quad F^\top F = I_C, \quad \sum_j S_{ij} = 1, \quad S_{ij} \geq 0 \text{ for all } (i, j),$$

$$\text{where} \quad L(S, F) = \mathbf{tr}(F^\top (I_n - S)F),$$

here $\lambda, \beta > 0$ are tuning parameters (see Appendix 7.3). $F$ is an auxiliary $C$-dimensional variable (with $C < n$) and is constrained to consist of orthogonal columns. $S$ is constrained to be a stochastic matrix and further regularized by the squared Frobenius norm, i.e. $||S||_F^2$.

In order to represent the resulting denoised network, the solution $S$ can be made symmetric by $(S + S^\top)/2$. In addition, the objective and constraints in (OPT1) ensure two key properties for $S$ to represent a denoised network:

*Property (1): $S$ complies well with the links in network $W$.*

The first term in the objective function of (OPT1) involves maximizing the Frobenius product of $S$ and $W$, i.e.,

$$-\mathbf{tr}(W^\top S) = -\sum_{i,j} W_{ij} S_{ij}.$$

so each link in $S$ is consistent with $W$. Taking the sum of the element-wise products allows $S$ to be invariant to scaling of $W$.

*Property (2): $S$ is low rank and conveys cluster structures.*

The term $L(S, F)$ in (OPT1) is an imposed graph regularization on $S$ so that it is embedded in a low-dimensional space spanned by $F$. To see this, first note that $(I_n - S)$ is the graph Laplacian of $S$ as the row sums (and column sums) of $S$ is 1. It can be shown that

$$L(S, F) = \mathbf{tr}(F^\top (I_n - S)F) = \sum_{i,j} ||f_i - f_j||_2^2 S_{ij},$$

where $f_i$ and $f_j$ are the $i$th and $j$th rows of $F$ respectively. Thus, each row of $F$ can be interpreted as a $C$-dimensional embedding of the corresponding node in the network. Here, $|| \cdot ||_2$ denotes the $\ell^2$-norm, so the minimization of $L(S, F)$ enforces link $S_{ij}$ to capture the Euclidean distance of node $i$ and node $j$ in the vector space spanned by $F$.

## 2.2 Learning from multi-resolution networks

The general graph denoising framework above can be easily extended to incorporate additional information. Suppose instead of a single observation $W$, we have $m$ noisy observations or representations of the underlying network $S$. Denote these observations as $W_1, ..., W_m$. We refer to this multi-resolution network as $W$, where each link in $W$ contains $m$ different ordered values. (This terminology is not only used to conveniently correspond to the Hi-C interaction maps at different resolutions, but it also helps to remind us that the noise in each network is not necessarily identical or stochastic.) A multi-resolution network consists of different representations of $S$ and provides more modeling power than a single-resolution network [32]. We can use this additional information to extend (OPT1) to the following optimization problem:

$$\text{minimize} \quad -\mathbf{tr}\left(\left(\sum\nolimits_\ell \alpha_\ell W_\ell\right)^\top S\right) + \lambda\, L(S, F) + \beta||S||_F^2 + \gamma P(\alpha) \qquad \text{(OPT2)}$$

$$\text{with respect to} \quad S \in \mathbb{R}^{n \times n}, \quad F \in \mathbb{R}^{n \times C}, \quad \alpha \in \mathbb{R}^m$$

$$\text{subject to} \quad F^\top F = I_C, \sum\nolimits_j S_{ij} = 1, \sum\nolimits_\ell \alpha_\ell = 1, S_{ij} \geq 0 \text{ for all } (i, j), \alpha_\ell \geq 0 \text{ for all } \ell.$$

$$\text{where} \quad L(S, F) = \mathbf{tr}(F^\top (I_n - S)F), \quad P(\alpha) = \sum\nolimits_\ell \alpha_\ell \log \alpha_\ell,$$

where $\lambda, \beta, \gamma > 0$ are tuning parameters (see Appendix 7.3). The vector $\alpha = [\alpha_1, ..., \alpha_m]^\top$ weights the $m$ observed networks $W_1, ..., W_m$ and needs to be learned from the data.

The modification to the first term in the objective in (OPT1) from that in (OPT2) allows $S$ to simultaneously conform with all of the networks according to their importance. To avoid overfitting with the weights or selecting a single noisy network, we regularize $\alpha$ via $P(\alpha)$ in the objective of (OPT2). In our application, we chose $P(\alpha)$ so that the entropy of $\alpha$ is high, but one may select other penalties for $P(\alpha)$. (e.g. L1 or L2 penalties)

While (OPT2) is non-convex with respect to all three variables $S, L, \alpha$, the problem is convex with respect to each variable conditional on fixing the other variables. Therefore, we apply an alternating convex optimization method to solve this tri-convex problem efficiently. The three optimization problems are solved iteratively until all the solutions converge. The following explains how each variable is initialized and updated.

(1) *Initialization.*

The variables $S$, $L$ and $\alpha$ are initialized as

$$\alpha^{(0)} = \frac{1}{m}\mathbf{1}_m, \quad S^{(0)} = \sum\nolimits_\ell \alpha_\ell^{(0)} W_\ell, \quad F^{(0)} = [v_1^{(0)}, ..., v_C^{(0)}]$$

where $\mathbf{1}_m$ is a length-$m$ vector of ones, i.e, $\mathbf{1}_m = [1, ..., 1]^\top$. The weight vector $\alpha$ is set to be a uniform vector to avoid bias, and $S$ is initialized to be the sum of the individual observed networks $W_i$ according to the initial weights. Finally, $F$ is initialized to be the top $C$ eigenvectors of $S$, denoted as $v_1^{(0)}, ..., v_C^{(0)}$.

(2) *Updating $S$ with fixed $F$ and $\alpha$.*

When we minimize the objective function only with respect to the similarity matrix $S$ in (OPT2), we can solve the equivalent problem:

$$\text{minimize} \quad -\sum\nolimits_{i,j}\left(\sum\nolimits_\ell \alpha_\ell\, (W_\ell)_{i,j} + \lambda(FF^\top)_{i,j}\right) S_{i,j} + \beta \sum\nolimits_{i,j} S_{i,j}^2 \qquad \text{(OPT3)}$$

$$\text{with respect to} \quad S \in \mathbb{R}^{n \times n}$$

$$\text{subject to} \quad \sum\nolimits_j S_{ij} = 1, \quad S_{ij} \geq 0 \text{ for all } (i, j).$$

This optimization problem is clearly convex because the objective is quadratic in $S_{i,j}$ and the constraints are all linear. We used the KKT conditions to solve for the updates of $S$. Details of the solution are provided in Appendix 7.1.

(3) *Updating F with fixed S and $\alpha$.* When we minimize the objective function only with respect to the similarity matrix $F$ in (OPT2), we can solve the equivalent problem:

$$\begin{aligned}
\text{minimize} \quad & \mathbf{tr}(F^\top(I_n - S)F) && \text{(OPT4)}\\
\text{with respect to} \quad & F \in \mathbb{R}^{n \times C}\\
\text{subject to} \quad & F^\top F = I_C.
\end{aligned}$$

This optimization problem can also be interpreted as solving the eigenvalue problem for $(S - I_n)$ because the trace of $F^\top(S - I_n)F$ is maximized when $F$ is a set of orthogonal bases of the eigen-space associated with the $C$ largest eigenvalues of $(S - I_n)$. We used standard numerical toolboxes in MATLAB to solve for the eigenvectors.

(4) *Updating $\alpha$ with fixed $F$ and $S$.*

Now treating $S$ and $F$ as parameters, the equivalent problem with respect to $\alpha$ becomes a simple linear programming problem:

$$\begin{aligned}
\text{minimize} \quad & -\sum_\ell \alpha_\ell \sum_{i,j} (W_\ell)_{i,j} S_{i,j} + \gamma \sum_\ell \alpha_\ell \log \alpha_\ell && \text{(OPT5)}\\
\text{with respect to} \quad & \alpha \in \mathbb{R}^m\\
\text{subject to} \quad & \sum_\ell \alpha_\ell = 1, \quad \alpha_\ell \geq 0 \text{ for all } \ell.
\end{aligned}$$

Using the optimality conditions, we derived a close-form solution for $\alpha_\ell$ for each $\ell$:

$$\alpha_\ell = \frac{\exp\left(\frac{\sum_{i,j}(W_\ell)_{i,j}S_{i,j}}{\gamma}\right)}{\sum_\ell \exp\left(\frac{\sum_{i,j}(W_\ell)_{i,j}S_{i,j}}{\gamma}\right)},$$

Details are provided in Appendix 7.2.

(5) *Termination.*

The alternating optimization terminates when all three variables $S$, $F$, and $\alpha$ converge. Even though alternating optimization techniques are widely-used heuristic approaches, the parameters converged in approximately 20 iterations in the applications we have considered.

## 2.3 Learning from multi-resolution networks and highly confidence links

Now suppose in addition to a multi-resolution network, we are given noiseless (or highly confident) information about the presence of certain links in the network. More formally, we are given a set $\mathcal{P}$, such that if a link $(i, j) \in \mathcal{P}$, then we know that it is almost surely a true positive link. If $(i, j) \notin \mathcal{P}$, then we only know that this link was unobserved, and have no information whether or not it is present or absent in the true denoised network.

In the applications we consider, there is typically a subset of nodes for which *all* of their incident links are unobserved. So if we consider a binary adjacency matrix on these nodes based on $\mathcal{P}$, a number of columns (or rows) will indeed have all missing values. Therefore, the only information we have about these nodes are their incident noisy links in the multi-resolution network.

The formulation in (OPT2) can easily incorporate the positive set $\mathcal{P}$. For each node $i$, we denote $\mathcal{P}_i = \{j : (i, j) \in \mathcal{P}\}$ additional parameters and formulate an extended optimization problem

$$\begin{aligned}
\text{minimize} \quad & -f(S) - \tau \mathbf{tr}\left(\left(\sum_\ell \alpha_\ell W_\ell\right)^\top S\right) + \lambda\, L(S, F) + \beta||S||_F^2 + \gamma P(\alpha) && \text{(OPT6)}\\
\text{with respect to} \quad & S \in \mathbb{R}^{n \times n}, \quad F \in \mathbb{R}^{n \times C}, \quad \alpha \in \mathbb{R}^m\\
\text{subject to} \quad & F^\top F = I_C, \sum_j S_{ij} = 1, \sum_\ell \alpha_\ell = 1, S_{ij} \geq 0 \text{ for all } (i,j), \alpha_\ell \geq 0 \text{ for all } \ell\\
\text{where} \quad & f(S) = \sum_{i=1}^n \frac{1}{|\mathcal{P}_i|} \sum_{j \in \mathcal{P}_i} S_{ij}, \; L(S, F) \text{ and } P(\alpha) \text{ follow from (OPT2).}
\end{aligned}$$

Notice that when applying alternating optimization to solve this problem, we can simply use the same approach used to solve (OPT2). The only change needed is to include $f(S)$ in the objective of (OPT3) in order to update $S$.

# 3 Implementation Details

## 3.1 How to Determine $C$

We provide an intuitive way to determine the number of communities, $C$, in our methods. The optimal value of $C$ should be close to the true number of communities in the network. One possible approach to discover the number of groups is to analyze the eigenvalues of the weight matrix and searching for a drop in the magnitude of the eigenvalue gaps. However, this approach is very sensitive to the noise in the weight matrix therefore can be unstable in a noisy networks. We use an alternative approach by analyzing eigenvectors of the network, similar to [27]. Consider a network with $C$ disjoint communities. It is well known that the eigenvectors of the network Laplacian form a full basis spanning the network subspace. Although presence of noise may cause this ideal case to fail, it can still shed light on community membership. Given a specific number of communities $C$, we aim to find an indication matrix $Z(R) = XR$, where $X \in \mathbb{R}^{n \times C}$ is the matrix of the top eigenvectors of the network Laplacian, and $R \in \mathbb{R}^{C \times C}$ is a rotation matrix. Denote $[M(R)]_i = \max_j [Z(R)]_{i,j}$. We search for $R$ such that it minimizes the following cost function

$$J(R) = \sum_{i,j} \frac{[Z(R)]_{i,j}^2}{[M(R)]_i^2}$$

Minimizing this cost function over all possible rotations will provide the best alignment with the canonical coordinate system. This is done using the gradient descent scheme [27]. Instead of taking the number of communities to be the one providing the minimal cost as in [27], we seek the number of communities that result in the largest drop in the value of $J(R)$.

## 3.2 Convergence Criterion

The proposed method is an iterative algorithm. It is important to determine a convergence criterion to stop the iterations. Our method adopts a well-defined approach to decide convergence has been reached. Similar to spectral clustering [3], we use *eigengap* to measure the convergence of our method. Eigengap is defined as follows:

$$eigengap(i) = \lambda_{i+1} - \lambda_i \tag{1}$$

where $\lambda_i$ is the $i$-th eigenvalue of the matrix $S$ where we sort the eigenvalues in ascending order ($\lambda_1 \leq \lambda_2 \leq \ldots \lambda_n$). For $C$ clusters, we use $eigengap(C) = \lambda_{C+1} - \lambda_C$.

The intuition behind eigengap is that, if a similarity includes $C$ perfectly strong clusters, then $eigengap(C)$ should be near zero (which was proved in [28]). Due to the low-rank constraint in our optimization framework, we seek a small value of $eigengap(C)$ for a good optimal value. We can set a stopping criterion for our method using $eigengap(C) < T$ for a small threshold, $T$. However, due to the noise reaching a small threshold cannot be guaranteed, therefore, a practical stopping criterion adopted by our method is when $eigengap(C)$ has stoped decreasing. In our experiments we have observed that, $eigengap(C)$ usually decreases for around 10 iterations and then remains stable.

# 4 Experiments

We apply the framework presented in (OPT6) to Hi-C and Capture-C data. As explained, detecting communities in these data has important scientific ramifications. Our denoising strategy can be part of the pipeline for discovering these communities. We evaluated our methods on real data and checked their robustness by adding additional noise and measuring performance.

For the real data, we started with a ground truth of domains previously identified in the GM12878 cell line chromosomes 14 and 21 [10] , filtered to only contain domains that do not have ambiguous boundaries or that overlap due to noise in the ground truth, and stitched these together. We ran our algorithm using data at 8 different resolutions (5 kb,10kb, 25kb, 50kb, 100kb, 250kb, 500kb, 1Mb). A heat map of the highest and lowest resolution of the simulated data for chromosome 21 can be seen in Figure 1.

Figure 2a shows a heat map of the denoised version of chromosome 14 using (OPT6). Below the heat map we show the ground truth blocks. 1) The baseline (Louvain algorithm [29]) was set to the clusters determined from the highest resolution Hi-C map (purple) . 2) The clustering improves after denoising this map using (OPT1) (orange). 3) Pooling data through the use of multi-resolution maps and (OPT2) further increases the size of the clusters. Finally 4) using the high confidence set, multi-resolution and (OPT6) (blue).

As mentioned earlier, in order to determine ground truth, we have chosen large disjoint blocks with low levels of noise. To test our algorithm in the presence of noise, we added distance dependent random noise to the network. We evaluated our performance by measuring the normalized mutual information (NMI) between the ground truth and the clusters resulting from the noisy data Figure 2b [30]. We see that while the NMI from the baseline falls rapidly the performance of our denoising algorithm stays relatively constant after a rapid (but significantly smaller) drop. Figure 2c shows the weights assigned to each resolution as noise is added. We see that the weight assigned to the highest resolution has a steep drop with a small amount of noise. This could partially explain the drop in the performance of baseline (which is computed from the high resolution data) in Figure 2b.

To validate the obtained denoised network by our method, we check 2 features of domains: 1) increased covariation in genomic signals such as histone marks inside domains compared to across domains and 2) the binding of the protein CTCF at the boundaries of the domains (see Appendix 7.4). We quantify covariation in genomic signals by focusing on 3 histone marks (H3K4ME1, H3K4ME3 and H3K27AC), and computing the correlation of these across all pairs of genomic regions, based on measurements of these histone marks from 75 individuals [33]. We then compare the ratio between covariants with-in domains and covariants between domains . A higher ratio indicates better coherence of biological signals within domains while larger dispersions of signals between domains, therefore implying better quality of the identified domains. Second, we inspect another key biological phenomena that is binding strength of transcription factor CTCF on boundary regions [31, 35]. It has been observed that, CTCF usually binds with boundary of domains in HiC data. This serves as another way to validate the correctness of identified domain boundaries, by checking the fraction of domain boundaries that contain CTCF. Figure 2d shows that our method produces a higher ratio of specific histone marks and CTCF binding than the baseline, indicating better ability to detect biologically meaningful boundaries.

In each experiment, we selected the number of communities $C$ for clustering based on the implementation details in Section 3. The best $C$ is highlighted in Figure 3a-b. The optimal $C$ coincided with the true number of clusters, indicating that the selection criteria was well-suited for the two datasets. Furthermore, as shown in Figure 3c-d the alternating optimization in (OPT6) converged within 20 iterations according to the criteria in Section 3 where the eigen-gaps stabilized quickly.

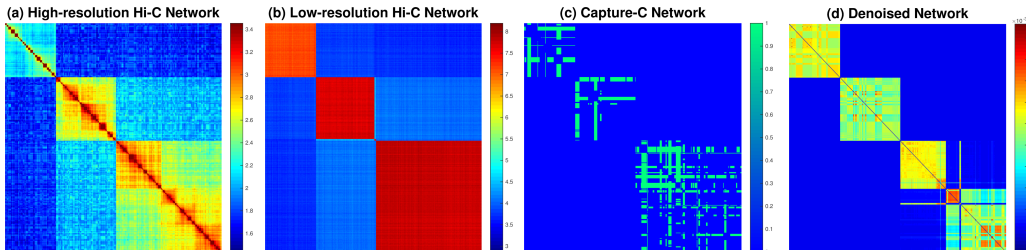

Figure 1: a) Heat map of data simulated from chromosome 21. The subclusters were chosen to be clearly distinguishable from each other (in order to have clear boundaries to determine the ground truth for the boundaries of the blocks). The blocks were subsequently stitched to each other. b) Simulated low resolution data. c) Capture-C network: these positions are treated as low-noise data. d) Denoised Network: (OPT6) was used to denoise the network using all 8 resolutions in addition to the Capture-C data in (c).

## 5  Conclusions and Future Work

In this paper we proposed an unsupervised optimization framework to learn meaningful structures in a noisy network. We leverage multi-resolution networks to improve the robustness to noise by automatically learning weights for different resolutions. In addition, our framework naturally extends to incorporate partial labels. We demonstrate the performance of our approach using genomic interaction networks generated by noisy Hi-C data. In particular, we show how incorporating multiple Hi-C resolutions enhances the effectiveness in denoising the interaction networks. Given partial information from Capture-C data, we further denoise the network and discover more accurate community structures.

In the future, it would be important to extend our method to whole genome Hi-C data to get a global view of the 3D structure of the genome. This will involve clever binning or partition of the genome to

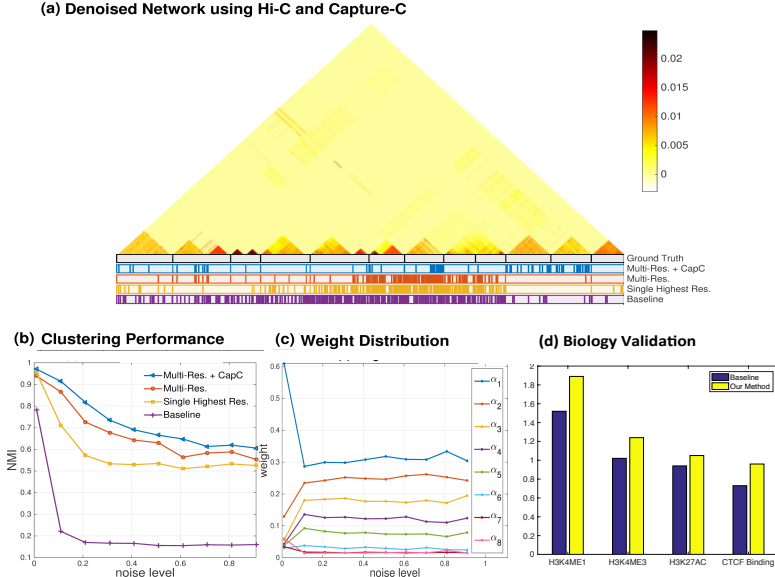

Figure 2: a) Denoised Network: heatmap of denoised network using with Hi-C and Capture-C according to (OPT6). The tracks below the heatmap indicate the division of the classified communities with respect to the ground truth. The use of multi-resolution Hi-C and capture-C achieves the best concordance with the ground truth. b) Clustering performance: The performance of the baseline degrades rapidly with the introduction of noise. Our method with various inputs perform significantly better than the baseline suggesting that denoising using our framework can significantly improve the task of clustering c) Weight Distribution: The weights ($\alpha_i$) assigned to each resolution from the optimization in (OPT2). The noise increases the performance of the highest resolution matrix decreases rapidly at first. In response, the method rapidly decreases the weight for this matrix. d) Ratio between covariates: we used three specific histone marks and the CTCF binding sites as indicators of the accuracy in detecting the boundaries.

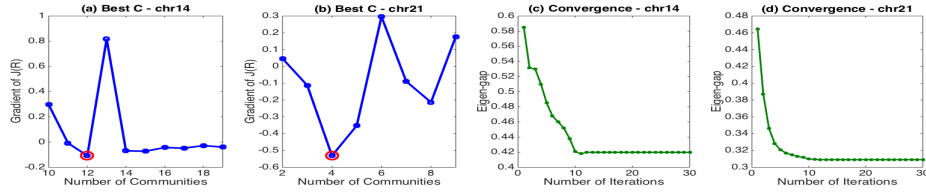

Figure 3: a) - b) The gradient of $J(R)$ over the number of communities $C$. The best $C$ selected is based on the value that minimizes the gradient of $J(R)$ (circled in red). c) - d) The eigen-gaps over the number of iterations in the optimization framework. The eigengaps stabilize at the same value within 20 iterations, indicating the optimization problem converges in only a few iterations.

reduce the problem size to a more local level where clustering methods can reveal meaningful local structures. In addition, our current framework is very modular. Even though we demonstrate our approach with k-means clustering a module, other appropriate clustering or community detection algorithms can be substituted for this module for whole genome Hi-C analysis. Finally, it would be interesting to extend our approach to a semi-supervised setting where a subset of confident links are used to train a classifier for the missing links in Capture-C data.

# 6 Acknowledgments

We would also like to thank Nasa Sinnott-Armstrong for initial advice on this project. JZ acknowledges support from Stanford Graduate Fellowship. AP was partially supported by Stanford Genome Training Program: NIH 5T32HG000044-17. AK was supported by the Alfred Sloan Foundation Fellowship. OU is supported by the HHMI International Students Research Fellowship. BW and SB were supported by NIH Sidow grant (1R01CA183904-01A1).

## Footnotes

*bowang87@stanford.edu

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
