[Supplementary Material · Appendix.pdf]

# 7 Appendix

## 7.1 Details for solving (OPT3)

Let $s_i \in \mathbb{R}^n$ denote the $i$th row of $S$, i.e, $S = [s_1, s_2, ..., s_n]^\top$, and define a $n \times n$ matrix $V = [v_1, v_2, ..., v_n]^\top$, where each $v_i \in \mathbb{R}^n$ for $i = 1, ..., n$, and the $j$th entry of $v_i$ is

$$(v_i)_j = -\frac{1}{2\beta} \left( \lambda(F^\top F)_{i,j} - \sum_\ell \alpha_\ell (W_\ell)_{i,j} \right),$$

for $j = 1, ..., n$. Notice that each $s_i$ is independent in the optimization problem (OPT3), so we can solve for $s_i$ using $u_i$ via a quadratic optimization subproblem

$$\begin{aligned} \text{minimize} \quad & \frac{1}{2}||s_i - v_i||_2^2 \\ \text{with respect to} \quad & s_i \in \mathbb{R}^n \\ \text{subject to} \quad & \mathbf{1}_n^\top (s_i)_j = 1, \quad (s_i)_j \geq 0 \text{ for all } j. \end{aligned} \tag{OPT7}$$

The constraints in each subproblem are simplex, so we can directly use the Lagrangian function

$$\mathcal{L}(\alpha, \delta, \sigma) = \frac{1}{2}||s_i - v_i||_2^2 - \delta(\mathbf{1}^\top s_i - 1) - \sigma^\top s_i,$$

where $\delta > 0$, and $\sigma_j > 0$ for all $j$, to derive the KKT conditions:

$$(s_i)_j - (v_i)_j - \delta - \sigma_j = 0, \ \mathbf{1}_n^\top s_i = 1, \ (s_i)_j \geq 0, \ \sigma_j \geq 0, \ (s_i)_j \sigma_j = 0, \text{ for } j = 1, ..., n.$$

Solving this system of equations, we get

$$(s_i)_j = \left( (v_i^*)_j - \sigma^* \right)_+, \ \sigma_j = \left( \sigma^* - (v_i^*)_j \right)_+,$$

where $(x)_+ = \max(x, 0)$,

$$v_i^* = \left( I_n - \frac{\mathbf{1}_n \mathbf{1}_n^\top}{n} \right) v_i + \frac{\mathbf{1}_n}{n} \quad \text{and} \quad \sigma^* = \frac{\mathbf{1}_n \sigma}{n}.$$

We can solve for $s_i$ directly be solving $\sigma^*$ by solving the following equation:

$$f(\sigma^*) = \frac{1}{n-1} \sum_{j=1}^{n-1} \left( \sigma^* - (v_i^*)_j \right)_+ - \sigma^* = 0.$$

Since $f(\sigma^*)$ is piecewise linear and convex, we can use Newton method to solve for $\sigma^*$.

## 7.2 Details for solving (OPT5)

We can use the Lagrangian function

$$\mathcal{L}(\alpha, \delta, \sigma) = -\sum_\ell \alpha_l \sum_{i,j} (W_\ell)_{i,j} S_{i,j} + \gamma \sum_\ell \alpha_\ell \log \alpha_\ell - \delta(\alpha^\top \mathbf{1} - 1) - \sigma^\top \alpha,$$

where $\delta > 0$, and $\sigma_\ell > 0$ for all $\ell$, to derive the optimality condition for each $\alpha_\ell$:

$$\frac{\partial \mathcal{L}(\alpha)}{\partial \alpha_\ell} = \sum_{i,j} (W_\ell)_{i,j} S_{i,j} - \gamma(1 + \log \alpha_\ell) + \delta + \sigma_\ell = 0.$$

Using the observation that

$$\alpha_\ell \propto \exp\left( \frac{\sum_{i,j} (W_\ell)_{i,j} S_{i,j}}{\gamma} \right),$$

along with the constraint $\sum_\ell \alpha_\ell = 1$, we can directly derive a closed-form solution

$$\alpha_\ell = \frac{\exp\left( \frac{\sum_{i,j} (W_\ell)_{i,j} S_{i,j}}{\gamma} \right)}{\sum_\ell \exp\left( \frac{\sum_{i,j} (W_\ell)_{i,j} S_{i,j}}{\gamma} \right)},$$

for $\ell = 1, ..., m$.

## 7.3 Parameter settings

It is important to determine the set of hyper-parameters $\beta$ and $\gamma$. We use a data-driven approach to adaptively learn these two hyper-parameters through the following rules:

$$\gamma = \beta = \frac{1}{N} \sum_{i=1}^N \sum_{j=1}^k (S_{i,k+1}^2 - S_{i,j}^2)$$

where $x_i$ denotes the $i$-th node, $x_i^j$ denotes the top $j$-th nearest neighbor of the $i$-th node and $k$ is a pre-defined parameters. We use $k = 10$ as default. Since these two parameters controls the regularization strength, we adaptively update the parameters during each iteration:

$$\gamma^{(t+1)} = \gamma^{(t)} * (1 + 0.5\delta\{eigengap(C) > 1e - 6\})$$

where $t$ denotes the current iteration number. This means that, we increase the value of $\gamma$ if $eigengap(C)$ is large.

To provide robustness to our method, we use a smoothed version of updating for $S$:

$$S^{(t+1)} = (1 - \alpha) * S^{(t)} + \alpha * S_+^{(t)}$$

where $S^{(t)}$ is the output of our method after $t$-th iteration and $S_+^{(t)}$ is the solution by solving OPT3 in the main text. Finally, we use $\alpha = 0.8$ as default.

## 7.4 Definition of validation metrics

Here we provide detailed metrics to evaluate the goodness of identified boundaries.

### 7.4.1 Normalized Mutual Information

Throughout the paper, we commonly use the Normalized Mutual Information (NMI) to evaluate the consistency between the obtained domains and the true labels of the $N$ regions. Given two clustering results $U$ and $V$ on a set of data points, NMI is defined as: $I(U, V) / \max\{H(U), H(V)\}$, where $I(U, V)$ is the mutual information between $U$ and $V$, and $H(U)$ represents the entropy of the clustering $U$.

Specifically, assuming that $U$ has $P$ clusters, and $V$ has $Q$ clusters, the mutual information is computed as follows:

$$I(U, V) = \sum_{p=1}^{P} \sum_{q=1}^{Q} \frac{|U_p \cap V_q|}{N} \log \frac{N|U_p \cap V_q|}{|U_p| \times |V_q|}$$

where $|U_p|$ and $|V_q|$ denote the cardinality of the $p$-th cluster in $U$ and the $q$-th cluster in $V$ respectively. The entropy of each cluster assignment is calculated by

$$H(U) = -\sum_{p=1}^{P} \frac{|U_p|}{N} \log \frac{|U_p|}{N},$$

and

$$H(V) = -\sum_{q=1}^{Q} \frac{|V_q|}{N} \log \frac{|V_q|}{N}.$$

NMI takes on values between 0 and 1, measuring the concordance of two clustering results. In the simulation, we calculated the obtained clustering with respect to the true labels. Therefore, a higher NMI refers to higher concordance with ground truth, i.e. a more accurate label assignment of each cell.

### 7.4.2 Data processing for HiC and Capture-C datasets

We focused this analysis on a subset of the genome, namely regulatory elements and their target genes. These were defined as continguous regions that fall in either enhancer or Tss chromatin states [34]. We used HiC data from [10] and Capture-C data from [35]. For HiC, we used the SQRTVC normalized values. To translate HiC to our regulatory elements, we assigned the maximum value from HiC across the bins that overlapped each element. For Capture-C, we used the reported promoter-promoter and promoter-other interactions. To translate Capture-C to our regulatory elements, we assigned the maximum value (log2 of observed over expected) obtained from the Capture-C.

### 7.4.3 Biology Validation Metrics

We also use some biology-related validation metrics to measure the quality of identified domains. First, based on previous studies [33], we expect variation in functional genomic data to be correlated inside domains but not across domains. To test this, we looked at data measuring histone marks across 75 individuals, which allowed us to compute pairwise correlations in the signal of H3K4ME1, H3K4ME3 and H3K27AC (in their respective peaks), across these individuals. Given a set of identified domains (denoted as $\{U_i, i = 1, 2, \cdots, q\}$, where $q$ is the number of domains) and these correlations across individuals between histone marks (denoted as $\{\mathbf{Cov} \in \mathbf{R}^{n \times n}\}$, where $n$ is the number of nodes), we define the ratio between covariants within domains and covariants between domains as follows:

$$R = \frac{1}{q} \sum_{i=1}^{q} \frac{\sum_{j \in U_i} \mathbf{Cov}(i, j) / |U_i|}{\sum_{j \in \widetilde{U_i}} \mathbf{Cov}(i, j) / |\widetilde{U_i}|}$$

where $\widetilde{U_i}$ denotes the nodes that is not in the set $U_i$. A higher value of $R$ indicates better coherences of nodes within domains in terms of histone mark covariants (correlations).

Second, we expect that domains have binding sites of the protein CTCF at their boundaries, so we can use this as a biological performance evaluation. To test this, we computed the percentage of domain boundaries that contain a CTCF binding site [31] in 2d for our method and for the baseline.