[Reviews · NeurIPS 2016]

Reviewer 1

Summary

The authors describe an optimization approach to identify clusters (communities) in a network while combining multiple noisy versions of the network. In addition their framework allows to incorporate high confidence evidence (though I’m not convinced that this part of their method is very effective). The authors apply their method to combine multiple resolutions of Hi-C data and analyze their approach on a simplified setting of disjoint blocks and in the presence of increasing amount of noise.

Qualitative Assessment

I believe the review of this paper should be done in 2 stages: 1) method; 2) application. The method, as presented, is fairly general and could be applied to many different scenarios. It is a relatively novel method for network de-noising – combining multiple networks from noisy observations of the true underlying networks, in particular network that is made of more or less clear clusters. In this context the method is well described. I would be interested to know how well does it scale – the complexity and running time of the method on networks of various size. I understand that it probably doesn’t scale too well given that the genome had to be broken up into chromosomes and the chromosomes that were presented were not the largest. What is the resolution? How many nodes was the biggest network tested? Regularization in section 2.3 (to add high confidence edges) is not really fixing the edges – you could have the same number of completely different edges and it would still look ok (since you are only looking at sums here). Can you check how much it hurts you? You could regularize the edges directly… I think it would be interesting to see the effects of this regularization (the results in the experimental section support the fact that you are not using your high-confidence edges to the fullest…) The application, I believe, is somewhat problematic. You start your paper talking about community detection. I’m not sure that’s the best or most important use of Hi-C data. You also keep talking about de-noising and community detection? Which is it? What is the biological motivation for community detection? Does one necessarily lead to the other? The link for me is missing. We don’t actually know whether true 3D genome data clusters nicely. Certainly, for many applications in biology, it would be nice to keep the 3D structure of this data preserved as a network rather than clusters. Going to the cluster level is really not the wholly grail of this data and in that, the method and the motivation of the work are misleading. In any case, I was not convinced by the motivation that cluster representation is the way to go (or even the right kind of embedding) for Hi-C data. The abstract was also misleading “we propose to mine useful structures from noisy networks”… - why are they useful? And also by structures here, I could imagine something more informative than relatively large clusters on which the method was ultimately tested. In the same spirit, I’m not sure that in the method, as applied in this particular scenario, S should be low ranked, sparse yes, but low rank? I’m not sure that is true of the biology here. Experiments To me, a more useful use of network denoising in the Hi-C data context would have been combining different resolutions of this data and testing the resulting denoised network with the Capture-C data – how well would your de-noising go? Also, there are other approaches combining multiple networks, such as Similarity Network Fusion (Wang et al, 2014) that this work could be compared to. When you describe figure 2c in the text, you shouldn’t really be describing what is on the figure – that goes in the caption. In the caption of the figure, you shouldn’t really be describing that one method is performing better than the other, just what the figure is about, not the conclusions one would draw from it. The experimental section definitely requires some clarification, it is very confusing now. In figure 2a it’s not clear what you used as a baseline. In figure 2b – why is the \alpha_8 coefficient also going down? Details: There are a lot of typos and strange use of long words, somebody has to read the manuscript very carefully and simplify/fix the English a bit. Section 3.1. Can you provide more intuition for J(R)? It’s not obvious what it captures. Line 123 L2 is not properly written In Figure 1 you first say that you combine (a,b,c) and then that you combine all 8 resolutions, so which is it? Please write what you actually have done and what is in the figure. If these are two different things, then please write that clearly. Figure 2c comes first, prior to Figures 2a and 2b, perhaps you could decouple these, so that they would be cited appropriately? They shouldn’t really be one figure

Confidence in this Review

3-Expert (read the paper in detail, know the area, quite certain of my opinion)


Reviewer 2

Summary

This work focuses on the analysis of Hi-C data (devoted to DNA sequence topology measures). To do that, it proposes an optimisation with constraints (or penalised add-ons) framework to reconstruct a network (OPT1 scheme). It is then modified to include multi-resolution data (OPT2) and then to include high-confidence edges in a semi-supervised framework (OPT6). The method is applied to data for which a Gold Standard is available.

Qualitative Assessment

Interesting method tailored for a specific data set, clearly motivated. Although the method can certainly be applied in many different scenarios. Since no theoretical results are available, maybe a simulation study to compare to state-of-the-art might be useful (not only same method without refinements). After all, your method is more an add-on in terms of penalised term to an existing optimisation scheme. Even if it is clearly stated. Some questions/remarks: - l44: aren't links from regulatory elements to targets? Explain which components are considered as sources/targets please. - end of 1.1/beg. 1.2: a bit disturbing that you motivate existing work limitations (concept is not even that clear) in 'Our Contribution' section. - l87-88: maybe cast the ability of including high-confidence edges in the semi-supervised literature? - l95-96: compared to what? - l99: better explain what the $ n $ elements in the network are, i.e. what the nodes represent. - l104: could you explain what it means that the network of interest S is low-rank please? - l111: important question here. Projection the current solution $ S $ on the space of symmetrical matrices is probably sub-optimal. Can you comment on potential pitfalls/solutions at least? (e.g. see Hiriart-Urruty and Lemaréchal, Convex analysis and minimization algorithms, 1993) - OPT2 optimisation scheme makes me think of multiple network inference (from multiple data sets) as in Chiquet et al. Statistics and Computing 2011. Some comments? Entropy $ P(\alpha) $ seems to favour using one source (depending on value for $ \gamma $). Comments? - l190: not sure the number of communities concept $ C $ has been introduced beforehand. - l194: not clear, compared to what? - l216: so no theoretical guarantees for the algorithm to converge right? - l259: computing time comparison? - End of conclusions: future work?

Confidence in this Review

2-Confident (read it all; understood it all reasonably well)


Reviewer 3

Summary

This paper presents a machine learning method to infer an interaction network and its community structure. The novelty of this method is that it is able to combine multiple noisy inputs at different resolutions and learn the underlying denoised network. The authors apply the method to denoising chromosome capture experiment data and calling chromosome contact domains from it. They demonstrate the potential application of this method in identifying genomic loci clusters with biological significance.

Qualitative Assessment

Overall, the article is well written and the intuition behind the method is well explained. However, the authors should address the following questions in the paper: 1) Bulk chromosome capture experiment data are typtically averages over the chromosome structures of multiple cell populations. One should expect that the community structure of the genomic loci interaction network is also a population average. This poses a challenge in partitioning the nodes into the correct community. For example, both Sij and Sik can be arbitrarily large, e.g., much larger than the average Savg, while Sjk is arbitrarily small, which means the Euclidean metric proposed in the low-dimensional embedding term L(S,F) might not be obeyed. 2) Capture-C exp. data from different samples are not directly comparable due to the unknown mapping from read count space to the intrinsic loci interaction space. A simple linear combination of different data sets doesn't always make sense. Some sort of normalization of the input matrices W is necessary, but this is never mentioned in the current version of the paper. 3) Fig. 1(d) shows two abrupt interaction bands in the lower right corner of the matrix between the 3rd and 4th mega-domains -- it reads as if a set of loci from a domain interact strongly with a small cluster of loci from another domain but do not at all interact with the loci up- and downstream of this cluster. One would expect the interaction intensity dies off somewhat gradually as a function of the genomic distance. This abrupt signal doesn't show up in the original data. Could this be an artifact of the denoising? 4) The authors generated a "ground truth" for benchmarking the performance of the method by "visually" extracting the clusters, but this is very subjective and it was never mentioned what visualization tools were used to aid this. How do we know, for example, if the algorithm is assigning boundaries (true positive) that the authors failed to identify? 5) More specific information regarding the performance of the method should be given. For example, judging from Fig 2(c), it appears that a lot of false positives appear in the prediction compared to the "ground truth" so it would helpful to plot the precision-recall curve as a function of the noise level to better understand the performance in the different settings. Also, since the "ground truth" identification is subjective, it might be more informative to compare the performance of this method to other methods in the literature. An alternative comparison would be to examine the overlap of the identified boundaries between different noise levels to see if the method can predict a consistent set of domains under different noise levels. Another issue is that the noise model used in the benchmarking and what we should exepct the identified domain boundary to change were not mentioned. 6) It might be helpful to compare the computational expense of this method in various settings. For example, how much more expensive it would be to combine all the possible inputs as compared to just a few?

Confidence in this Review

2-Confident (read it all; understood it all reasonably well)


Reviewer 4

Summary

The manuscript presented a method for community detections from noisy network data from Hi-C data. The method integrates multi-resolution interaction networks and high confidence partial labels from Capture-C data.

Qualitative Assessment

Firstly, there isn't sufficient comparisons against existing approaches give that community detections being a relatively mature field. Secondly, the experiments doesn't support the generality of the method. The author only tested on their method on one dataset, and it's unclear how representative the dataset is. Overall, I don't think the present manuscript provides enough evidence to be considered for publication.

Confidence in this Review

2-Confident (read it all; understood it all reasonably well)


Reviewer 5

Summary

In this paper, the authors present an unsupervised learning framework which denoises complex networks. The framework consists on several optimization problems, introduced incrementally, from the simplest case to the most complex. More concretely, the authors introduce modifications to handle multi-resolution networks and partial labels. An explanation of how the optimizations are implemented is provided in every case. The authors validate empirically their method by conducting community detection experiments on Hi-C and capture-C data from chromosomes.

Qualitative Assessment

The paper constitutes a very complete piece of research and a pleasure to read. The authors present their problem, they explain thoroughly their solution reaching every detail and they validate experimentally their methodology, obtaining obviously satisfying results. The method they present is able to tackle several in principle complicated aspects of the problem while still being easy to understand and implement. The authors describe their methodology very thoroughly but incrementally, so that it is straightforward to follow and the author does not get lost. They explain every aspect of the optimization framework, making their experiments highly replicable. The experimental results, conducted on Hi-C and Capture-C data of chromosomes, are very strong and presented in a graphical way, very easy to interpret. After reading through all the paper, I have nothing but some minor comments to the authors, in order to make the paper even clearer: -I must have missed the reason why S is a doubly stochastic matrix, as you state in Line 121. You might refer to the resulting matrix of applying the symmetrization technique described in Line 111. If this is the case, please, change the notation to avoid confusions with the S obtained as a result of (OPT1), and explain why the properties 1 and 2 are not affected by this symmetrization. In another case, please, clarify. I also suppose that you perform this symmetrization in the more complex cases ((OPT2) and (OPT6)). Please, clarify this, as well. -What does the N in Line 162 mean? Why is it different from n? -I suppose that the summation over j in the formula after Line 209 starts at 1. Please, make it explicit or correct me. In addition, in Line 211 you state that both parameters beta and gamma are adaptively updated. Does beta follow the same updating rule as gamma? -Using alpha for the weight of the smooth updating of S is very confusing, as alpha has been used throughout the paper as the vector of weights of the observed networks. I would pick another letter. -I suppose that when you mention (OPT7) in Line 249 it is a typo and it is really referring to (OPT6), as no (OPT7) is defined. Please, correct it. -Your experiments have the goal to detect disjoint communities. Out of curiosity, have you thought of how your methodology would behave in case of having overlapping communities? Do you think the results would be as accurate as the ones you present for disjoint communities? This would be a very interesting extension of your work. -A minor linguistic revision of the paper is encouraged.

Confidence in this Review

3-Expert (read the paper in detail, know the area, quite certain of my opinion)


Reviewer 6

Summary

The contribution of this paper is to develop a method which integrates experimental data from multiple resolution chromatin conformation maps to result in a better quality interaction network. The method first compiles information from several data sources into a single incidence matrix on which additional methods for denoising could then be applied.

Qualitative Assessment

To my understanding, the authors develop a method for combining interaction data from multiple datasets that contain differing amounts of noise and resolution information through alternating optimization of S, F, and alpha conditioned on W1-Wm in which S and W1-Wm are the incidence matrices and the goal is to maximize the Frobenius product of S and W [to enforce consistency of the denoised network S with several datasets W1-Wm (each with learned weighting parameter alpha_1-alpha_m)] in conjunction with regularization with an auxiliary matrix F [to enforce S is low rank]. The authors test the performance of this method on data derived from multi-scale chromosome conformation data from a lymphoblastoid human cell line that has been visually annotated to recover ground truth clusters and then introduce additional noise, which shows good performance. More thorough analysis of the expected noise properties and ground truth for the chromatin conformation data would be helpful to know how well the method will generalize to other experimental datasets.

Confidence in this Review

2-Confident (read it all; understood it all reasonably well)